# Climate and the Parasite Paradox: Tick–Host Networks Depend on Gradients of Environmental Overlap

**DOI:** 10.3390/pathogens14101025

**Published:** 2025-10-10

**Authors:** Agustín Estrada-Peña

**Affiliations:** 1Faculty of Veterinary Medicine, University of Zaragoza, Miguel Servet 177, 50013 Zaragoza, Spain; antricola@me.com; 2Ministry of Human Health, CSAI-Foundation, 28-014 Madrid, Spain

**Keywords:** tick–host relationships, niche overlap, climate gradient

## Abstract

This study investigates how climate gradients shape tick–host associations, testing the hypothesis that variations in climate leverage some associations, which can be ecosystem-specific. To test this hypothesis, we modelled tick–host associations across the Western Palearctic using climatic variables and a large dataset of georeferenced tick (seven species, *n* = 23,462) and vertebrate host records (*n* = 6.5 million across 162 species aggregated into 50 genera). Niche overlap with hosts is highly variable but consistently significant (*p* < 0.05) in every tested ecosystem of the target territory. Montane grasslands exhibit the lowest values of tick–host niche overlap, meaning that they support the smallest but still resilient set of available hosts. Host phylogenetic diversity (PD) depends on the ecosystem rather than tick species; PD is lowest in montane grasslands (supporting previous results) and in the case of *D. reticulatus* in savannas and scrubland. Nestedness of tick–host networks, known to be related to the resilience of parasite–host networks, is highest in climatically extreme ecosystems, reflecting adaptability of tick–host networks, as measured by niche overlap on modelled distribution. Multidimensional scaling confirms that host community composition and niche overlap vary significantly across ecosystems, supporting the hypothesis of host rewiring under diverse climatic conditions. These findings may have important implications for the concept of community composition and the circulation of tick-borne pathogens.

## 1. Introduction

Parasitism is notably conservative in host range, showing high host specificity at ecological and evolutionary levels [1]. Parasites are believed to become more adapted to their hosts over time, explaining host–parasite relationships evolution [2,3]. Ecological fitting provides a strong explanatory framework, contrasting with the idea that current species associations result from coevolution, where species exert mutual selective pressures [4,5]. Although coevolution is often seen as the main mechanism shaping these associations, distinguishing it from ecological fitting is challenging, as ecological fitting allows only ‘fitting’ associations to persist without direct coevolution [6,7,8,9,10].

Ecological fitting is the process by which organisms adapt to new environments, resources, or interactions using traits evolved under past conditions for new purposes [11]. In host–parasite associations, ecological fitting can occur through mechanisms like resource tracking, where a parasite exploits the same resource type in a different host or environment. This aligns with Hutchinson’s multidimensional niche concept [12], where a species can expand its range or niche position if conditions sufficiently match its fundamental needs.

Resource tracking explains the “parasite paradox,” where parasites, despite being specialists, shift to new hosts over time, with sister species surviving on each other’s hosts even without natural co-occurrence [12]. Ecological fitting requires a species to encounter a new environment or host and achieve fitness using traits evolved in a different context [13]. Generally, true niche shifts in parasites are rare, with gradual adaptation to new abiotic variables being more common [14,15,16]. The limits of ecological fitting and the genetic traits involved are largely unknown.

Tick–vertebrate associations offer a unique system to test hypotheses about tick–host relationships under environmental changes, niche conservatism, and biotic interactions. The niche overlap among ticks and hosts raises questions about whether climate gradients create high-contact areas that could aid pathogen persistence. It is known that ticks and hosts form associations through ecological specialization in generalist ticks, rather than coevolution [17]. In general terms this seems to be in contrast with other parasites [9,10]. It has been demonstrated that the range of hosts (the vertebrate community composition) may affect pathogen circulation, which relies on competent vertebrate reservoirs in which ticks feed [18,19]. Niche conservatism, where species retain ancestral traits, suggests ticks occupy the same fundamental niche across regions, independent of host composition [17]. However, modelling is necessary to study these relationships because data may be sparse for large regions. Thus, it is hypothesized that the community composition of reservoir hosts is responsible of the density of tick-borne pathogens.

Studying vertebrate–parasite associations benefits from tools like interaction networks and climate suitability modelling. Networks offer a graphical and quantitative framework for understanding organism relationships, while climate models reveal ecological patterns like niche overlap and shared climatic resource use by ticks and hosts [19,20]. Together, these tools provide insights into parasitic system dynamics, niche availability, and niche overlap with hosts. This, and because hosts are a suitable system to explore with these tools. Many ticks have interest for human and/or animal health, and their survival and reproduction strongly depend on the climate and the niche overlap between the parasites and the hosts.

This study aims to show that climate gradients influence the tick–host relationships. We hypothesize that, using a modelling approach based on a large dataset of field derived data, changes in the niche overlap between ticks and hosts could be pinpointed in a variety of ecosystems. Using data for the Western Palearctic, we aimed to demonstrate that the community of hosts “available” for ticks is dependent upon the climate. Our goal is to illustrate that climate gradients significantly shape tick–host associations, offering an ecological explanation for the variation in host use across different environments.

## 2. Materials and Methods

### 2.1. Aim

We employed correlative species distribution modelling based on climatic variables to analyze a database of georeferenced records of ticks and vertebrates across a target area spanning from 26° W, 71° N (upper left corner) to 57° E, 30° N (lower right corner). This territory was analyzed at a spatial resolution of 4 km per pixel. An existing classification of ecosystems was used as an ecological descriptor of the study area, providing the framework for subsequent comparisons.

To evaluate niche overlap between ticks and vertebrates within each ecosystem, we constructed coincidence matrices of parasite–host overlap, which were then used to generate co-occurrence networks for each ecosystem. We analyzed the composition of these networks to assess whether ticks exhibit similar phylogenetic diversity and host composition across regions, or whether the structure of tick–host associations changes substantially between ecosystems. To test for niche divergence (versus niche conservatism) and for nestedness within the contact networks, we applied a set of ecological indices.

### 2.2. Sources of Data

A comprehensive dataset of tick records in the Western Palearctic was published and made publicly available [21]. For the present study, we used distributional data for eight tick species: *Dermacentor marginatus*, *Dermacentor reticulatus*, *Haemaphysalis punctata*, *Hyalomma lusitanicum*, *Hyalomma marginatum*, *Ixodes ricinus*, *Rhipicephalus annulatus*, and *Rhipicephalus bursa*. The dataset comprises 23,462 georeferenced records, of which approximately 75% have a spatial resolution of ~4 km and the remainder a resolution of about 0.5 km. Other tick species known to occur in the target area were excluded owing to insufficient records for reliable modelling, or because of their endophilic behaviour which renders them largely independent of climatic gradients and confined to niches that cannot be reliably mapped or modelled.

Vertebrate records were obtained from the same compilation. We used data on 162 vertebrate species reported as hosts of the focal ticks, aggregated into 50 genera to minimize redundancy. To construct a reliable dataset of vertebrate hosts, we consulted [21], which documents vertebrate hosts of ticks over the past 40 years. Additional host data were retrieved from the Global Mammal Parasites Database [22] and updated with records from GBIF (https://gbif.org; see reference [23] for a list for individual downloads and accession dates). Duplicate records were identified and removed. In total, the vertebrate dataset comprised 6,538,201 records. While the dataset was compiled at species level, subsequent analyses were conducted at genus level to manage the logistical challenges of modelling such a large dataset.

We calculated the phylogenetic diversity of hosts, for each tick, at each ecological region. This is an important feature because it is not a simple count of species, but the calculation of diversity based on a phylogenetic tree. The calculation of the phylogenetic diversity of hosts for ticks for the ecosystems studied was performed using a synthetic phylogenetic tree, obtained from the Open Tree of Life initiative (http://opentreeoflife.org, accessed on 16 August 2024) using the rotl package v3.1.0 [24] in R [25]. The Open Tree of Life provides a synthetic view of the phylogeny of life based on published data and genetic repositories (see https://tree.opentreeoflife.org/for details, accessed on 8 September 2024).

Climatic data used to model environmental suitability for ticks and vertebrates were obtained from the TerraClimate repository (https://www.climatologylab.org/terraclimate.html, accessed on 7 March 2023 [26]). Monthly data for maximum temperature, minimum temperature, and vapour pressure deficit were compiled and averaged across the target area for the period 1990–2020 using the terra package in R. Harmonic regression was performed on the monthly averages per year and the first three coefficients of each regression were retained as explanatory variables [27,28]. This yielded nine climatic predictors of three coefficients each for maximum temperature, minimum temperature, and vapour pressure deficit. A map of the ecological regions considered in this study, together with the main climatic features of the study area, is shown in Figure 1. The script used to calculate harmonic regression is provided as Appendix A.

### 2.3. Modelling

We employed a stacked species distribution ensemble modelling (SSDM) approach with multiple algorithms to select the most effective model for each taxon. This approach allows us to capture potential interactions between vertebrates and their non-linear associations with climate variables [29]. We utilized the SSDM package [30] for R and employed the algorithms GAM, MARS, and SVM. For GAM, we conducted 500 iterations per species/genus with an epsilon value of 1 × 10^−8^. For MARS, we allowed up to 4 degrees of interactions. For SVM, we maintained the same epsilon value as for GAM and performed 3 cross-validation steps. We kept all other parameters at their default values. In all cases, 70% of the records were used for model training, while 30% were used for model testing. The software automatically generated pseudo-absences across the target territory, ensuring a one-to-one correspondence with the organism records. The metrics used to select the best models for inclusion in the ensemble SDM were values of sensitivity and specificity, omission rate, and correctly detected records, together with the Cohen’s kappa. The maps displaying the modelled distribution of the target ticks are shown in Figure 2. All the raw maps are available in Appendix A, together with the metrics of every model.

### 2.4. Measures of Niche Sharing Among Ticks and Vertebrates and Phylogenetic Diversity

We assessed niche occupancy using principal components analysis (PCA) combined with the convex hull method, which together provide a robust framework for quantifying environmental niches [14]. PCA reduces correlated climatic variables into orthogonal axes (principal components) that capture maximum variance in reduced dimensions. Habitat overlap between ticks and vertebrate genera was then quantified in this PCA-derived environmental space, using hypervolume concepts and convex hulls.

The convex hull defines the smallest convex polygon enclosing all occurrence points in the PCA space, representing a species’ realized niche breadth. The area of the hull approximates niche size, while overlap among hulls reflects the degree of niche similarity or divergence. The R script used for PCA and niche overlap calculations is provided as Appendix A.

Suitable climate niche areas were estimated for each tick species and vertebrate genus within ecosystems defined by the Classification of Terrestrial Ecosystems (https://www.worldwildlife.org/publications/terrestrial-ecoregions-of-the-world, accessed 4 January 2022). Ecosystems considered included tundra, temperate grasslands, temperate forests, montane grasslands, Mediterranean woodlands, boreal forests, and temperate conifer forests. For each ecosystem, we calculated the percentage of niche overlap between each tick species and vertebrate genera in the PCA-reduced space. Phylogenetic diversity of vertebrate hosts was quantified for each region using Pagel’s λ [31]

To evaluate variation in tick–host associations among ecosystems, we applied multidimensional scaling (MDS). MDS visualizes similarities among objects in a distance matrix, here representing the inverse of habitat-sharing percentages between ticks and hosts across ecosystems [32]. Objects are positioned in low-dimensional Cartesian space, so that between-object distances approximate the original distances as closely as possible. In this study, clustering of points in MDS space indicates biogeographical structuring and stable host associations across environments, whereas separation suggests altered proportions of niche overlap, network rewiring, and changes in host community composition. Stress values below 0.05 were taken as strong evidence of a reliable dimensional reduction.

### 2.5. Networks Construction

We built directed networks (ticks -> host) connecting interacting taxa by links whose weight corresponds to the shared habitat in the reduced space. Metrics describing network structure have frequently been associated with measures of network stability or resilience [33]. We concentrated on two metrics of network structure that have been previously linked to network stability: nestedness and modularity. Nestedness is a property of ecological networks where species interactions form a hierarchical structure in which the interactions of less-specialized species are a subset of those of more-specialized species [34,35]. This concept holds particular significance in the study of parasitic networks, as it often reflects the degree of specialization of species. Modularity represents the propensity for groups of nodes to engage in numerous interactions among themselves and exhibit sparse connections to other groups of highly connected nodes [36]. Modularity in ecological networks is commonly utilized to describe functional groups within ecosystems.

## 3. Results

Figure 3 presents the abiotic (climate) niche for the six target tick species, along with the similarity of niche with the niche of available hosts (biotic suitability): the latter is quantified as the percentage of the shared niche between each tick and its host evaluated with the Jaccard’s test of similarity among niches. An abiotic suitability below 20% falls below the threshold necessary for population persistence and is invariably associated with a significantly low sharing of niche with hosts. Two species, namely D. reticulatus and H. lusitanicum, have a reduced suitability in several territories. The Mediterranean-type ecosystem exhibits the highest abiotic suitability for all species, likely due to the greater diversity of ticks colonizing Mediterranean ecosystems. Ixodes ricinus stands out as the species that colonizes the widest range of ecosystems.

Host availability, measured as the niche overlap among ticks and hosts, is very variable, with significance values below 0.05 in all instances (Mantel test). Values of Jaccard’s test of niche similarity are lowest for every tick species in the montane grassland ecosystem for every combination of tick and ecosystem. This indicates an incomplete niche sharing, where each tick species is loosely linked to available hosts but not entirely dependent on a single or a few hosts. Notably, *I. ricinus* is the sole tick capable of finding low but suitable abiotic conditions in both tundra and taiga ecosystems, in both instances because of substantial niche sharing with available hosts. We interpret this finding as a flexible switch of hosts as they become available under the diverse climatic conditions in the studied ecosystems.

The phylogenetic diversity of hosts for ticks is primarily driven by the ecosystem rather than the tick species (Figure 4). To note, this index is not related to the abundance of hosts but to their diversity in each ecosystem. PD values are lowest in montane grassland and scrubland, supporting the findings on the low niche similarity among ticks and hosts in that ecosystem. The association of *D. reticulatus* + savannas and scrubland also has a low PD of hosts. For the rest of ticks and ecosystems, PD has medium to high values, demonstrating that other than different hosts in each ecosystem ticks have a large PD available, demonstrating a high diversity of hosts. These findings point to an almost invariable PD accessible to ticks in all but a few ecosystems.

Figure 5 illustrates the values of nestedness in the interactions among ticks and hosts in each ecosystem. This index is interpreted as the tendency for nodes in the network to contact with subsets of interacting partners, and reflects resilience of the network. To note, the nestedness is a measure of the relationships among organisms in each ecosystem, and it is not dependent of one single organism. All the values are highly significant, according to Mantel test (*p* < 0.05 for every instance); maximum and highly significant values have been found for ecosystems with extreme climates, like taiga and tundra. We interpret these findings as the tendency of ticks to interact with groups of hosts that also share large environmental niches among them, a strategy aimed to maximize the hosts availability even in areas with weak support for tick survival.

Multidimensional scaling (MDS) further corroborated the previous findings. MDS summarizes the similarity of the values of niche sharing between ticks and hosts into a two-dimensional construct that helps to interpret these similarities. Figure 6 shows the results of MDS carried out on the similarity of host community composition at the resolution of each ecosystem. All the results have *p* < 0.05, supporting that the biotic suitability (or the host community composition and the niche overlap with ticks) is significantly different in each ecosystem, thus corroborating the hypothesis of the rewiring of host networks in each ecosystem. If ticks have similar host composition in each ecosystem, points in the chart would appear closer; however, MDS demonstrated that host sharing among ticks depends on the ecosystem, and sustained previous results of this study, like the similarity of the host composition for I. ricinus in ecosystems like tundra and boreal regions (i.e., closer in the MDS). Temperate and Mediterranean-type sites have similar associations (e.g., a minimal rewiring of the hosts networks) while savannas have a variable costs composition according to tick species.

## 4. Discussion

The role of vertebrates as hosts or reservoirs for tick-borne pathogens has been extensively studied (see [21] for a comprehensive review). Literature has explored host–tick–habitat interactions in natural settings (e.g., [37,38]) and broader ecological dynamics (e.g., [39]). Empirical research has linked tick abundance to host density and community composition (e.g., [40,41]), though these relationships often vary regionally. The dilution effect hypothesis suggests that increased biodiversity, through a balanced vertebrate community, can reduce tick-borne pathogen prevalence (e.g., [42]).

In this study, we analyzed a large dataset of records of ticks, hosts, and climate across the Western Palearctic. Our aim was to explore whether niche similarity and host associations vary systematically with climate gradients. We conducted a series of strict tests, including niche similarities, nestedness of resulting networks, and ordination using multivariate methods. The results revealed that several tick species colonizing a range of ecosystems possess and may adapt to a flexible spectrum of available hosts. In the studied system, tick–host relationships are not tick-specific but rather ecosystem-derived. Consequently, the ability of a community of vertebrates and ticks to circulate a pathogen, derived from the community composition, would be climate-driven. This is an important finding about the impact of the climate trends on tick-borne pathogens that needs to be demonstrated over large regions.

By quantifying niche similarity, phylogenetic diversity, and tick–host network resilience, we demonstrated that tick–host interactions are flexible and context-dependent. Ticks should therefore not be considered strictly limited by specific vertebrate hosts. Their distributions are primarily determined by climatic conditions, while tick–host networks change in response to local community composition. This ecological flexibility explains their epidemiological resilience and highlights the necessity of integrating both abiotic and biotic dimensions into predictive models of vector-borne disease. Notably, some results provide the limits over which ticks could not adapt to new conditions and would fail to establish a network of host contacts; for example, this applies to specific ecosystems, like the Mediterranean shrubs, which seem to be unsuitable for the colonization of the studied species. Further on this, species like *H. lusitanicum* or *D. reticulatus* seem to strictly adhere to subsets of either abiotic or biotic conditions, demonstrating an adaptability to the conditions existing in portions of the target territory (e.g., [21]).

These findings indicate that tick populations may have a flexible adaptation to climate, with their range being relatively independent of strict host specificity, except for monoxenous species. This has been particularly demonstrated for the coldest or driest ecosystems in this study as ticks share niches with other vertebrates absent in other ecosystems, giving rise to new associations. We tested the resilience of these networks, which were proven to be similar to tick–host associations observed in other ecosystems. This may be of relative importance for the tick life cycle, but it is of significance for tick-borne pathogens: unique combinations of ticks and vertebrates could promote unique patterns of pathogens circulation. Tick-borne pathogens exploit specific configurations of tick and host networks, an argument most studied for *Borrelia* spp. or *Anaplasma phagocytophilum* (e.g., [43,44,45,46,47]). Spirochaetes in the *B. burgdorferi* s.l. group are known to be speciated due to their associations with specific vertebrate groups (e.g., [43,44]). This epidemiological feature left a footprint in the spatial distribution of *Borrelia* spp. in the Western Palearctic, with its phylogenetic footprint still trackable [48].

The emergence and re-emergence of infectious diseases—especially vector-borne ones—have garnered increasing attention over the past two decades, largely because of climate change and habitat modification on pathogen-transmission dynamics [49]. Tick-borne pathogens are highly responsive to such environmental shifts and are especially sensitive to climate variability [49,50]. Rising temperatures and altered climatic regimes drive shifts in tick geographic ranges and seasonal activity (phenology), thereby enhancing opportunities for transmission and, in some cases, enabling host-switching without requiring evolutionary adaptation [51,52]. It has been hypothesized that parasites may encounter and exploit novel hosts through ecological fitting, facilitated by altered abiotic conditions [53].

The dynamic process of host changes in a parasite system cannot be solely modelled on the few dimensions of climate. However, results suggest that climate influences the likelihood of niche overlap between parasites and vectors, resulting in a diverse range of hosts under varying abiotic conditions. This is supported by multiple independent lines of evidence: (a) niche similarity analyses revealed significant overlap between ticks and hosts, but low congruence across ecosystems; (b) high phylogenetic diversity of hosts was consistently observed for nearly all tick species under different environmental conditions; and (c) multidimensional scaling (MDS) demonstrated that vertebrate community composition is structured by ecosystem type rather than tick identity. These findings confirm that ticks can opportunistically switch hosts across abiotic settings, maximizing their potential for persistence and influencing the circulation of specific pathogens.

Our results partially address the so-called “parasite paradox” as although ticks may appear host-specific at local scales, broader spatial analyses reveal a generalist strategy constrained primarily by abiotic factors. Apparent specialization often reflects ecological filtering imposed by local vertebrate communities rather than intrinsic host fidelity. These findings support theoretical predictions [14] and underscore the importance of studying host–parasite associations across environmental gradients, looking for phylogenetic signatures between parasites and vertebrates, or exploring the impact of climate in the existing associations. No solid assessment of the impact of climate change on human health and tick-borne pathogens can be carried out without a parallel rating of its impact on tick hosts.

## 5. Conclusions

Contrary to some recent hypotheses, our analysis does not support the notion that climate change alone will inevitably expand the range of tick-borne pathogens. Rather, climate change is expected to shift the geographic distribution of ticks, thereby exposing them to novel vertebrate communities which may, or may not, contribute to circulating these pathogens in new territories. Whether these communities contain competent reservoirs will ultimately determine transmission outcomes. Hence, climate-driven expansion will lead not to a simple broadening of host range, but to a restructuring of host–vector–pathogen networks.

While large-scale vector distribution models based solely on abiotic variables may be adequate, accurate prediction of pathogen risk requires incorporating vertebrate community data to reflect the true complexity of host competence. Future research should refine this approach by integrating near real-time, region-specific datasets—such as NDVI (normalized difference vegetation index) and temperature—to enhance our capacity to anticipate shifting epidemiological landscapes in response to global change.

## Figures and Tables

**Figure 1 pathogens-14-01025-f001:**
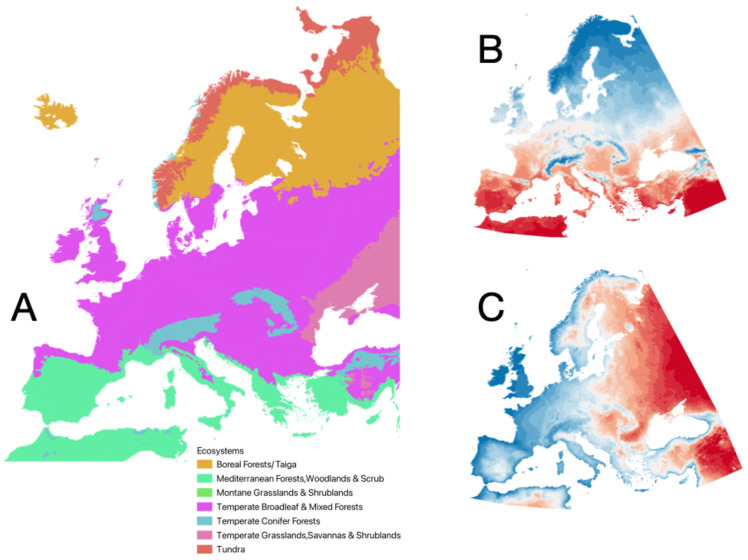
The spatial distribution of the ecosystems in the target territory (**A**). The average annual temperature in the target region (**B**), redder is warmer. The gradient of continentality (**C**), more blue is more Atlantic (contrary to continental).

**Figure 2 pathogens-14-01025-f002:**
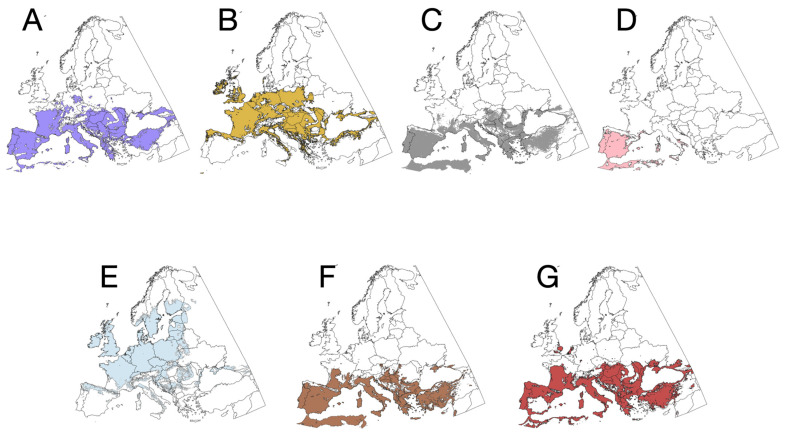
The projected binary distribution (presence/absence) of the six species of ticks used in this study. (**A**) *Dermacentor marginatus*. (**B**) *Dermacentor reticulatus*. (**C**) *Hyalomma marginatum*. (**D**) *Hyalomma lusitanicum*. (**E**) *Ixodes ricinus*. (**F**) *Rhipicephalus bursa*. (**G**) *Haemaphysalis punctata*.

**Figure 3 pathogens-14-01025-f003:**
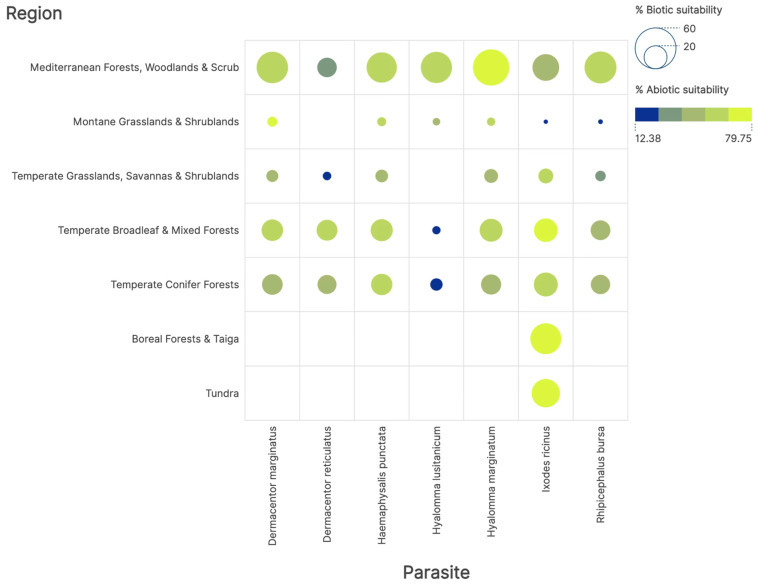
The relationships between tick species, ecosystems, percentage of abiotic (climate), and biotic (hosts) suitability. The size of each dot is proportional to the biotic suitability as the average Jaccard’s index of similarity of the niches of each tick with all the hosts colonizing the same ecosystem (in the range 0–100), and the colours reflect the abiotic suitability. Values of abiotic suitability below 20% could be below the threshold required for population persistence.

**Figure 4 pathogens-14-01025-f004:**
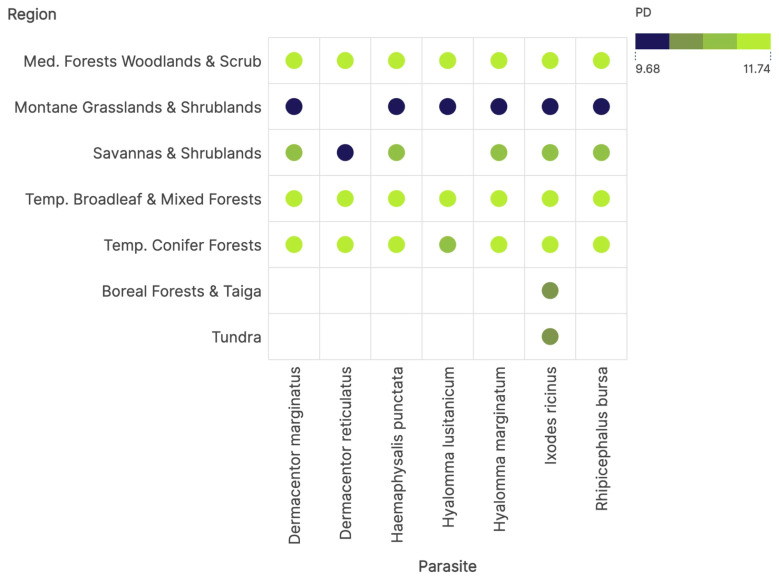
The values of phylogenetic diversity of the networks of ticks and hosts in each ecosystem.

**Figure 5 pathogens-14-01025-f005:**
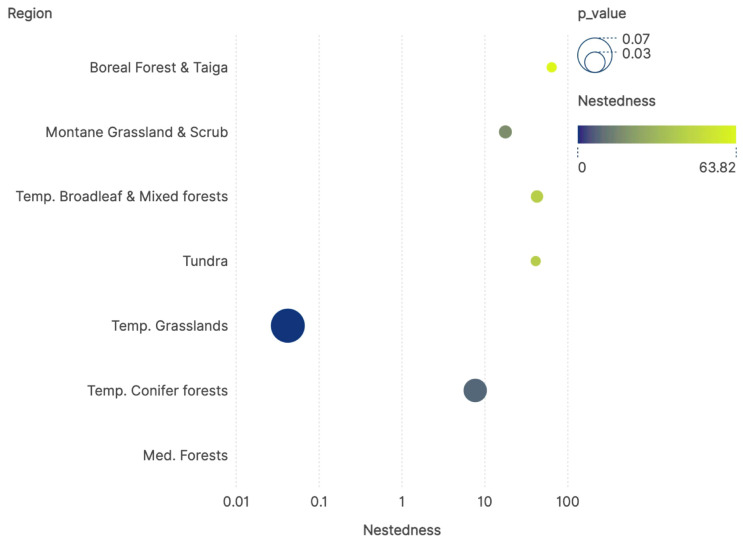
The values of nestedness of the network of ticks and hosts in each ecosystem, together with the value of the Mantel test for differences among ecosystems which states the significance of the nestedness value.

**Figure 6 pathogens-14-01025-f006:**
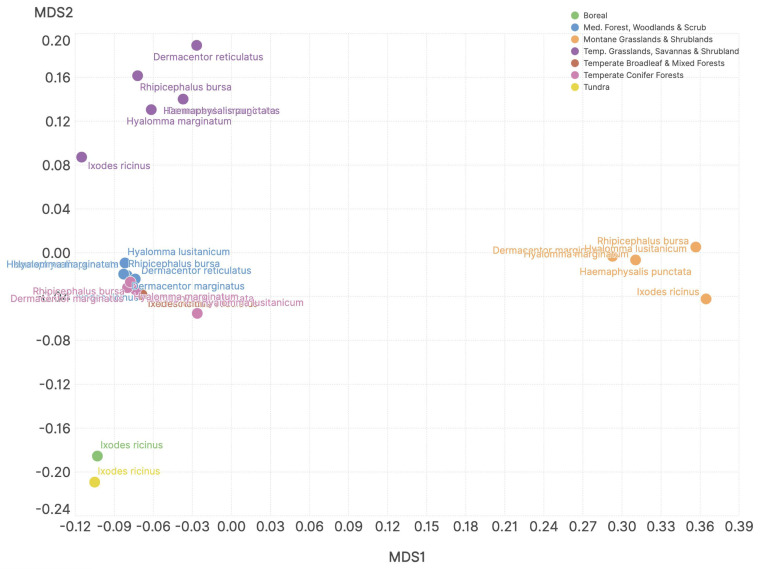
Multidimensional scaling (MDS) of the similarity of niche sharing values among ticks and hosts in each ecosystem. Symbols represent the relative position in the reduced space of the niche sharing between ticks and hosts in each ecosystem (dots with the same symbol refer to the same ecosystem). Colours show species of ticks. The relative distance among sites or among ticks are indicative of higher or lower differences in the niche sharing and therefore of the hosts composition, driven by the ecosystem (they appear closer) but not by the host preferences of the tick species (ticks are separated in the MDS). Colours representing the ecosystems are close to each other; symbols representing tick species are more separated among them, suggesting that relationships between ticks and hosts are different.

## Data Availability

Complete raw data and R script for harmonic regression of climate are available in the FigShare repository with the DOI 10.6084/m9.figshare.30053980.

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
