# Peer review of "Climate and the Parasite Paradox: Tick–Host Networks Depend on Gradients of Environmental Overlap"

_pathogens, 2025, doi:10.3390/pathogens14101025_

Round 1
Reviewer 1 Report
Comments and Suggestions for Authors
The article "Climate and the parasite paradox: ticks rewire the networks of hosts in large gradients of climate" is an interesting and well-prepared manuscript addressing a significant problem from epidemiological perspectives.
Author Response
Reviewer #1
The article "Climate and the parasite paradox: ticks rewire the networks of hosts in large gradients of climate" is an interesting and well-prepared manuscript addressing a significant problem from epidemiological perspectives.
Thank you very much for your comments. We prepared a new, improved version of the paper, that we hope is better explained in some parts.
Reviewer 2 Report
Comments and Suggestions for Authors
Dear Authors,
Your article “Climate and the parasite paradox: ticks rewire the networks of hosts in large gradients of climate” is a valuable contribution to the scientific discussion on the effects of climate change on tick distribution and the emergence of tick-borne pathogens in new areas. I believe that its publication in Pathogens will help clarify the expected changes in tick species occurrence in a changing world, shaped not only by climate change but also by host species rearrangements driven by anthropogenic factors.
My only concern is that some readers of Pathogens may be unfamiliar with certain ecological or modeling terms used in the manuscript. I think it would be helpful if you included brief explanations of these terms, as you already do in some cases. For instance, terms such as ecological fitting, niche conservatism, and niche specialization are explained in the text, which is very convenient. I believe it would be equally useful to provide clarification for other terms, such as phylogenetic signature, cascade failure, continental (as used in the context of Figure 1), and others.
Furthermore:
In Figure 1, it is very difficult to distinguish the colors of “Mediterranean Forests, etc.” and “Montane Grasslands, etc.” If possible, please use a different color scheme.
It is not clear why some regions appear differently across figures. For example, Temperate Grasslands do not appear in Figure 4; Mediterranean Forests, Woodlands and Scrub do not appear in Figure 5; and only six regions are shown in Figure 6.
Author Response
Reviewer #2
Your article “Climate and the parasite paradox: ticks rewire the networks of hosts in large gradients of climate” is a valuable contribution to the scientific discussion on the effects of climate change on tick distribution and the emergence of tick-borne pathogens in new areas. I believe that its publication in Pathogens will help clarify the expected changes in tick species occurrence in a changing world, shaped not only by climate change but also by host species rearrangements driven by anthropogenic factors.
My only concern is that some readers of Pathogens may be unfamiliar with certain ecological or modeling terms used in the manuscript. I think it would be helpful if you included brief explanations of these terms, as you already do in some cases. For instance, terms such as ecological fitting, niche conservatism, and niche specialization are explained in the text, which is very convenient. I believe it would be equally useful to provide clarification for other terms, such as phylogenetic signature, cascade failure, continental (as used in the context of Figure 1), and others.
Thank you very much for your comments and for the improvement of the paper. We agree that this is completely necessary in a context where many readers want to read but may not be experts on the topic. It is important to transmit information with adequate explanations where necessary. Most of the paper has been revamped: the introduction and discussion are new sections, the methods have been improved, and the results section is now shorter. We hope you will find this version better readable.
Furthermore:
In Figure 1, it is very difficult to distinguish the colors of “Mediterranean Forests, etc.” and “Montane Grasslands, etc.” If possible, please use a different color scheme.
It is not clear why some regions appear differently across figures. For example, Temperate Grasslands do not appear in Figure 4; Mediterranean Forests, Woodlands and Scrub do not appear in Figure 5; and only six regions are shown in Figure 6.
Thank you. This was produced due to a poor selection of colours that looked correct at real size but were deeply affected after being reduced to fit figures into the template. Additionally, there was a mistake when preparing some figures. To address these issues, we have remade all the figures (except for figure 1) with a new set of colours that allow for greater divergence between the minimum and maximum values. We have also redrawn all the figures using the corrected set of categories. We hope this new look will meet the standards of quality expected by the journal.
Reviewer 3 Report
Comments and Suggestions for Authors
This manuscript addresses the question: how climate gradients influence tick–host associations and whether such associations can be described as ecosystem-specific “rewiring.” The authors combine species distribution modelling, phylogenetic diversity analyses, and network metrics to explore these interactions using a very large dataset of ticks and vertebrate hosts across the Western Palearctic. The integration of correlative niche modelling with ecological network approaches is innovative, and the results have implications both for ecological theory (ecological fitting, niche conservatism, parasite paradox) and for applied issues (pathogen circulation under climate change).
Major comments
The paper attempts to address multiple broad questions simultaneously: ecological fitting vs. coevolution, niche conservatism vs. divergence, host “rewiring,” network resilience, and implications for pathogen transmission. It is therefore difficult to delineate what precisely the hypothesis is raised and tested.
The aggregation of hosts at the genus level may obscure meaningful ecological and epidemiological variation, since host competence often varies dramatically within a genus. It will be good to explicitly discuss this limitation and, if possible, test whether results hold when using species-level data for well-sampled taxa.
The study relies on correlative models, which are excellent for describing associations but limited for inferring causation. The claim about “rewiring” or adaptive flexibility needs, to my opinion, to be tempered, since no temporal or experimental data are included.
While the results show differences in niche overlap and network structure across ecosystems, the leap to pathogen circulation and epidemiological consequences remain speculative. No pathogen prevalence or transmission data were included. The authors should either (a) clearly delimit these points as hypotheses for future work.
The term “rewiring” implies dynamic processes, but the data are essentially static (spatial correlations). The authors should clarify what they mean by “rewiring” and acknowledge the limits of their evidence.
I found that parts of the discussion revert to general ecological theory (e.g., ecological fitting, coevolution) without always linking these concepts directly to the presented results. The discussion would be stronger if more tightly grounded in the specific findings (e.g., low niche overlap in montane grasslands, high nestedness in tundra/taiga).
Minor comments
Stylistic: The manuscript is quite long; tightening the introduction and discussion would make the central message clearer.
This is a promising and ambitious study with a strong dataset, but I recommend major revision to sharpen the framing, address methodological limitations, and temper some of the broader claims.
Author Response
Reviewer #3
This manuscript addresses the question: how climate gradients influence tick–host associations and whether such associations can be described as ecosystem-specific “rewiring.” The authors combine species distribution modelling, phylogenetic diversity analyses, and network metrics to explore these interactions using a very large dataset of ticks and vertebrate hosts across the Western Palearctic. The integration of correlative niche modelling with ecological network approaches is innovative, and the results have implications both for ecological theory (ecological fitting, niche conservatism, parasite paradox) and for applied issues (pathogen circulation under climate change).
Major comments
The paper attempts to address multiple broad questions simultaneously: ecological fitting vs. coevolution, niche conservatism vs. divergence, host “rewiring,” network resilience, and implications for pathogen transmission. It is therefore difficult to delineate what precisely the hypothesis is raised and tested.
The aggregation of hosts at the genus level may obscure meaningful ecological and epidemiological variation, since host competence often varies dramatically within a genus. It will be good to explicitly discuss this limitation and, if possible, test whether results hold when using species-level data for well-sampled taxa.
The study relies on correlative models, which are excellent for describing associations but limited for inferring causation. The claim about “rewiring” or adaptive flexibility needs, to my opinion, to be tempered, since no temporal or experimental data are included.
While the results show differences in niche overlap and network structure across ecosystems, the leap to pathogen circulation and epidemiological consequences remain speculative. No pathogen prevalence or transmission data were included. The authors should either (a) clearly delimit these points as hypotheses for future work.
Thank you for your comments. After reading your comments about the mansurcipt and re-reading the text, we noticed that some parts are over-stated, and, as mentioned by the reviewer there is a subtle trend to claim some effects that were not demonstrated completely by the results. Following the recommendations by the Reviewer we (1) wrote new sections for both Introduction and Discussion, that now state clearly the problem, are shorter, and keep the topic “within” the facts that have been demonstrated, (2) it is explicitly stated in these sections that results come from models, and that therefore, our results are to be tested in the field. We hope this new approach will be better readable.
The term “rewiring” implies dynamic processes, but the data are essentially static (spatial correlations). The authors should clarify what they mean by “rewiring” and acknowledge the limits of their evidence.
Thank you for your comments. We acknowledge this was a poor choice of words. We removed every comment about “rewire” in the text and replaced by sentences that do not transmit the impression about a dynamic process. Some concepts have been removed, and some others, reformulated. In any case, we tried to adhere to a standard
I found that parts of the discussion revert to general ecological theory (e.g., ecological fitting, coevolution) without always linking these concepts directly to the presented results. The discussion would be stronger if more tightly grounded in the specific findings (e.g., low niche overlap in montane grasslands, high nestedness in tundra/taiga).
Thank you for your comments. We addressed these topics aiming for a round Discussion, addressing the results and focusing on the most important parts of the manuscript.
Minor comments
Stylistic: The manuscript is quite long; tightening the introduction and discussion would make the central message clearer.
Thank you for your comments. As mentioned above, we managed to reduce pats of the manuscript writing new introduction and discussion, but also addressing other changes in both methods and results. Th manuscript is now shorter and we think it reads better and faster.
This is a promising and ambitious study with a strong dataset, but I recommend major revision to sharpen the framing, address methodological limitations, and temper some of the broader claims.
Done as requested. Thank you.
Round 2
Reviewer 3 Report
Comments and Suggestions for Authors
Thank you for taking my comments into account